# Multifunctional Polypyrrole-Based Textile Sensors for Integration into Personal Protection Equipment

**DOI:** 10.3390/s24051387

**Published:** 2024-02-21

**Authors:** Carolin Gleissner, Paul Mayer, Thomas Bechtold, Tung Pham

**Affiliations:** Research Institute of Textile Chemistry and Textile Physics, University of Innsbruck, Hoechsterstrasse 73, 6850 Dornbirn, Austria; carolin.gleissner@uibk.ac.at (C.G.); paul.mayer@uibk.ac.at (P.M.); thomas.bechtold@uibk.ac.at (T.B.)

**Keywords:** multifunctional sensor, temperature, ammonia, hydrogen chloride, polypyrrole

## Abstract

Integrated safety sensors for personal protection equipment increasingly attract research activities as there is a high need for workers in delicate situations to be physically monitored in order to avoid accidents. In this work, we present a simple approach to generate thin, homogeneous polypyrrole (PPy) layers on flexible textile polyamide fabrics. PPy layers of 0.5–1 µm were deposited on the fabric, which thus kept its flexibility. The conductive layers are multifunctional and can act as temperature and gas sensors for the detection of corrosive gases such as HCl and NH_3_. Using three examples of life-threatening environments, we were able to monitor temperature, atmospheric NH_3_ and HCl within critical ranges, i.e., 100 to 400 ppm for ammonia and 20 to 100 ppm for HCl. In the presence of HCl, a decrease in resistance was observed, while gaseous NH_3_ led to an increase in resistance. The sensor signal thus allows for distinguishing between these two gases and indicating critical concentrations. The simple and cheap manufacturing of such PPy sensors is of substantial interest for the future design of multifunction functional sensors in protective clothing.

## 1. Introduction

Smart textiles attract increasing interest to be used in the personal protection equipment sector. Very often, the working conditions of rescue forces, as well as of workers who operate with hazardous material, must be monitored by appropriate sensory equipment to prevent accidents. As a result of workers’ stress in high-risk situations, their body temperature or heart rate can increase rapidly [1]. Moreover, the workers’ senses can be tricked due to higher adrenaline in extreme situations. There are cases of firefighters who suffer from burns during a fire without feeling the heat, thus exposing them too long to the heat [2,3]. Smart textiles contain sensors that can detect the physical or chemical properties of the environment and create a signal. In the case of personal protective equipment, this signal can trigger an alarm to warn the person of life-threatening situations [4]. In the last few years, we have utilised conductive-coated textiles as sensors for different applications, e.g., strain sensors [5], multipoint temperature arrays and thermoelectric generators [6] based on the metal coating approach. One of the major issues of metal coating remains the limited flexibility of the coated textiles. As an alternative, intrinsically conductive polymers, e.g., polypyrrole, polyaniline and poly-3,4-ethylendioxythiophen, are increasingly receiving attention for conductive coatings [7]. Polypyrrole (PPy) is an abundantly used conductive polymer, extensively studied for its sensing properties. PPy-based sensors can be used for measuring temperature [8], detecting DNA and working as immunosensors [9]. Also, sensing properties for gases and vapours have been reported, e.g., ozone [10], ammonia [11], nitrogen dioxide [12,13], hydrogen [14,15], carbon dioxide [16,17]. PPy can be deposited on substrates via a lift-off process [11], electrochemical deposition [10,15], UV photopolymerisation [18], vapour deposition polymerisation [19] or chemical polymerisation [16,20].

Different conductive organic polymers have been proposed for the detection of gaseous NH_3_, e.g., PPy, PPy/BN and PPy/dodecylbenzene sulfonic acid/boron nitride [21]. In a recently published work, fungal chitin grafted polyaniline was tested as sensor material [22]. PPy-based sensors have also been studied as highly sensitive electrochemical sensors for NH_3_ and H_2_S in aqueous medium [23]. 

The design of such sensors often involves a high level of technical and equipment expenditure. In contrast, we chose a simple synthesis with low expenditure on equipment to generate thin, homogeneous PPy layers. The synthesis consists of the immersion of the fabric in an aqueous PPy solution and subsequently in an aqueous sodium persulfate solution. The simplicity of the process is especially important for textile processing to ensure a continuous operation with a high throughput. PPy is mainly deposited on smooth surfaces such as foils, glass, etc. In this study, we used flexible polyamide 66 (PA66) fabrics to integrate the sensing area directly into the textile with the aim of maintaining its flexibility and wearing comfort. A textile construction shows a bigger accessible surface compared to foils or glass with the same sample dimension (e.g., length × width) for volatile molecules. Therefore, higher adsorption and desorption capability can be ensured for the textile compared to smooth surfaces, which can lead to better sensor performance. A PA66-based textile was chosen as this material exhibits excellent mechanical performance in terms of strength and abrasion resistance, as well as good wear comfort. Due to this high level of performance, PA66-based fabric is frequently used for technical applications as well as for garment production [24]. The main goal of the study is to demonstrate the potential of flexible PPy-coated PA66 fabrics in personal protection equipment to warn of three dangerous sources within life-threatening ranges: heat, ammonia and hydrogen chloride gas. The temperature was monitored within a range between 25 and 70 °C. Large amounts of ammonia are used as a cooling agent in refrigerant plants, the chemical industry and fertilizer production. The concentrations of HCl and NH_3_ in the test protocols were selected with regard to critical concentrations that already cause serious health risks. Concentrations between 20 and 30 ppm can cause mild irritations. Higher concentrations of ammonia exhaust due to unforeseen events can cause skin irritation and mucosa destruction in workers [20]. Concentrations above approximately 1700 ppm can be deadly [25]. For this reason, we monitored ammonia concentrations in the air in a range between 100 and 400 ppm. Causes for the release of hydrogen chloride can be found, for example, in the production of chemicals, food processing or in the textile industry [26]. Inhalation of HCl above 3 ppm can cause irritation of the mucous membranes, particularly in the respiratory tract and eyes. A value of 50 ppm was determined as an IDLH value (immediately dangerous to life or health) [27]. Therefore, we investigated HCl concentration in air within a range between 20 and 100 ppm. The results presented in this work indicate a strategy to prepare flexible textile-based sensors for gaseous HCl and NH_3_. The response of the sensors was characterised by model systems. The cheap and simple technique to produce such sensors will permit their disposable design, which also will allow for regular replacement, e.g., after use under harsh environmental conditions.

## 2. Experimental

### 2.1. Materials

The PA66 fabric with a mass per unit area of 80 g m^−2^ and a yarn count of 110 dtex was kindly provided by Getzner Textil AG, Austria, used as received and washed. PA66 squares with dimensions of 3.5 cm × 5 cm were cut out. For the polymerisation, pyrrole (C_4_H_5_N, 98%) and sodium peroxodisulfate (Na_2_S_2_O_8_, ≥98%) was supplied by Merck KGaA, Darmstadt, Germany. All chemicals were used without further purification.

### 2.2. Polymerisation of Polypyrrole on PA66 Fabrics

Four PA66 fabrics were immersed in 0.04 M pyrrole solution in deionised water for 60 min at room temperature under constant agitation. Aqueous Na_2_S_2_O_8_ solution (molar ratio 1:1.1) was added, and the fabrics were exposed to the solution for another 60 min under constant agitation [28]. The volume of the pyrrole solution corresponds to the volume of the Na_2_S_2_O_8_ solution. The fabrics were washed extensively with deionised water.

### 2.3. Analysis of Polypyrrole Coated Polyamide 66 Fabrics

#### 2.3.1. Scanning Electron Microscopy (SEM)

Scanning electron images were taken using the benchtop Hitachi TM 4000, Hitachi Ltd. Corporation, Tokyo, Japan, with an acceleration voltage of 15 kV. The samples were sputtered with gold before analysis.

#### 2.3.2. Fourier Transform Infrared Spectroscopy (FTIR)

The attenuated total reflectance was recorded on the PA66 fabric surface in the spectral range of 4000 to 600 cm^−1^ using an FTIR spectrometer (Invenio FTIR Spectrometer, Bruker Optik GmbH, Ettlingen, Germany) equipped with an ATR unit from PIKE Technologies (Madison, WI, USA) with a resolution of 2 cm^−1^ and 64 scans per measurement.

#### 2.3.3. Flexural Rigidity Measurement

The flexural rigidity was determined at room temperature with the stiffness tester (Model 112, TABER Industries, North Tonawanda, NY, USA). For the test of the textile samples in our study, procedure BS 3356:1990 was slightly modified. Due to the small size of the PA66 fabrics, a weight of 0.352 g was attached to the end of the fabric prior to measurement. The sample overhanging length *L*_o_ in cm was determined at an angle of 41.5° (Figure 1). The bending length *L*_b_ (in cm) is defined as half of the overhanging length *L*_o_.

The flexural rigidity *G* in mg*·*cm was calculated by the weight per unit area *W* in mg*·*cm^−2^ and *L*_b_ (Equation (1)). The result is presented as a mean of three measurements.
(1)$$G=W\cdot L_{b}^{3}=W\cdot {\left(\frac{L_{o}}{2}\right)}^{3}$$

#### 2.3.4. Electrical Resistance Measurements

The electrical resistance measurements were carried out with the Keysight U 1733C resistance metre from Keysight Technologies, Santa Rosa, CA, USA.

### 2.4. Performance Tests

#### 2.4.1. Temperature Measurement

The sample was attached to the multi-meter with adhesive copper tapes and clamps and then exposed to a change in temperature of a heating plate. The resistance was continuously recorded during temperature changes.

#### 2.4.2. Exposure to Hydrogen Chloride and Ammonia Gas

The HCl and NH_3_ atmosphere was generated using HCl and NH_3_ solutions of three concentrations that are able to generate a defined vapour pressure of 20, 50 and 100 ppm HCl or 100, 200 and 400 ppm NH_3_ in the atmosphere according to [20,21]. For the sensor testing, mass fractions were considered; thus, 1 ppm is equal to 1 mg HCl or NH_3_ per 1 kg air. The vapour pressure was calculated as follows (Equations (2)–(4)):(2)$$p=\frac{n_{HCl/NH_{3}}\cdot R\cdot T}{V_{air}}with\;R=8.31\frac{J}{K\cdot mol}\;T=293\;K$$
(3)$$n_{HCl}=\frac{m_{HCl/NH_{3}}}{M_{HCl/NH_{3}}}$$
(4)$$V_{Air}=\frac{m_{Air}}{\rho_{Air}}=\frac{1\;kg}{1204\;kg/m^{3}}$$

The vapour pressure as a function of the concentration of HCl and NH_3_ in solution is shown in Table 1 and Table 2 [29,30]:

The PPy-coated PA66 fabrics were cut into stripes with 0.5 cm width. A copper film coated with conductive adhesive was used to contact the sensor textile. The dimensions of the accessible sensor surface area were 3 cm × 0.5 cm, and the contact area between the sensor and the copper tape was 1 cm × 0.5 cm on each side. Due to the high resistance of the sensor element (MOhm) and the large contact area, the resistance between the copper stripe and sensor fabric can be neglected. Adhesive tapes were used to cover the copper tape to avoid any corrosion effects (Figure 2a). A schematic drawing of the sensor is given in Figure 2c. The sensor was inserted over the bottle lid and connected to the multi-meter. The sensor was brought into contact with the liquid in the bottle (40 mL solution per 70 mL flask (Figure 2b,d)).

The experimental setup of the tests was designed to demonstrate the functionality of the sensor at different concentrations of HCl and NH_3_ in principle. In future technical development, calibration with standard measuring equipment will be required to define the range of applications and the accuracy of the sensing elements better.

## 3. Results and Discussion

### 3.1. Characterisation of Polypyrrole Layer

#### 3.1.1. Microscopy and Layer Thickness

After the polymerisation, the PPy-coated samples were analysed via SEM. The top views of pristine PA66 fabrics and PPy-coated ones are shown in Figure 3. SEM pictures do not show distinct differences between the fabrics, as a thin layer of PPy had been deposited. The PPy coating can be observed, in particular, on fibres that are positioned near the surface of the fabric. In addition, minor PPy deposits are detected in the free areas of the fabric. This particulate material was likely deposited from the solution during the phase of fibre coating with PPy (Figure 3, marked in red). To investigate the layer thickness of PPy, cross-sections of PPy-coated PA66 fabrics were analysed (Figure 4). A layer thickness between 0.5 and 1 μm was determined on a fibre with an approximate diameter of 22 μm. This demonstrates the coating of PA66 fibres with a thin PPy layer, which had been formed as a result of the polymerisation process.

#### 3.1.2. FTIR Analysis

The characteristic IR absorption bands of PA66 with their assignments can be observed for untreated fibres at 3300 cm^−1^ (N–H stretching), 1634 cm^−1^ (C=O stretching), 1534 cm^−1^ (C–N stretching and C–N–H deformation) and 1275 cm^−1^ (C–N stretching and N–H in-plane bending) [33] (Figure 5). The characteristic IR absorption bands of PPy are observed at 3120 cm^−1^ attributed to N–H stretching, 1556 cm^−1^ assigned to C=C stretching, 1470 cm^−1^ assigned to C–N stretching and 1042 cm^−1^ attributed to C–C out-of-plane vibration [34,35]. The PPy-coated PA66 fabric shows the characteristic IR absorption bands of PA66. In addition, new peaks at 1042 cm^−1^ (C–H_ip_ deformation), 962 cm^−1^ and 790 cm^−1^ (C–H out-of-plane bending for α-α’ connection of pyrrole rings) that correspond to PPy are also present (marked in grey) [36,37]. These results confirm a successful deposition of PPy on PA66 fabrics.

#### 3.1.3. Flexural Rigidity

The flexural rigidity of untreated PA66 fabric and PPy-coated PA66 fabric were analysed with a stiffness tester. For the untreated fabric, a value of 28.1 ± 5.8 g cm was determined, while for the PPy-coated fabric, a value of 30.3 ± 3.3 g cm was obtained (Figure 6). Regarding the standard deviations, hardly any change in value due to the treatment was determined. Often, the deposition on fabrics can occur on individual fibres as well as between them. The fibres in the fabric may stick together, forming a fused layer [38]. As a result, the fabric loses its flexibility and increases its resistance against bending. In our case, hardly any change was determined after PPy coating. This might be due to the deposition of a very thin PPy layer on the individual fibres, which is confirmed by a layer thickness below 1 µm in SEM images (Figure 4). In addition, the sticking together of fibres on cross-sections and top-view of fabric was not observed (Figure 3 and Figure 4). The observations on SEM images and rigidity measurements are in alignment. It can be concluded that the deposition of PPy leads to a thin PPy layer on PA66, and thus, the flexibility of the fabric changes only slightly. As a result, PPy coating of PA66 fabrics is a suitable technique for the manufacturing of flexible textile sensors.

### 3.2. Polypyrrole-Coated Polyamide 66 Fabrics as Multifunctional Sensing Surfaces

#### 3.2.1. Polypyrrole-Coated Polyamide 66 Fabric as Temperature Sensor

The temperature dependence of the textile sensor was characterised according to Section 2.4.1. With PPy-coated PA66 fabrics, a decrease in resistance with increasing temperature is observed. This process is reversible. When the temperature decreases, the resistance increases. It is assumed that an increase in temperature leads to more electrons being able to overcome the band gap barrier of PPy. As a result, a larger number of electrons are available for charge transport [8]. In addition, higher temperatures can lead to an increase in conjugation length in the polymer and can favour electron delocalisation, which also contributes to an increase in conductance [39]. The resistance follows Equation (5) with ρ, ρ_0_, *T*, *T*_0_ and *D* as the resistivity, the resistivity at infinite temperature, the temperature, characteristic temperature and the number of dimensions, respectively [8]:(5)$$\rho =\rho_{0}\cdot {\exp\left(\frac{T_{0}}{T}\right)}^{\frac{1}{1+D}}$$

Within a temperature range of 25 to 70 °C, an approximately linear correlation was observed for our sensor sample. For instance, from the evaluation of the first heating curve, a linear approximation according to Equation (6) with a coefficient of correlation R^2^ over 0.99 was obtained.
*y* = −0.0192*x* + 2.0733(6)

The cooling curves, as well as the heating and cooling curves of the second and third cycles, also show a linear behaviour (Figure 7). However, the intercept and slope differ slightly compared to the curve of the first heating cycle. This deviation might be due to contacting problems between the sensing surface and copper tape.

#### 3.2.2. Polypyrrole Coated Polyamide 66 Fabrics as Hydrogen Chloride Gas Sensor

The response of conductivity as a function of HCl concentration can be assigned to a doping effect of the polymer. The nitrogen atom of PPy becomes protonated by HCl, which induces a positive charge of the polymer backbone. As a result, cations are generated, which increase the conductivity by being mobile across the polymer. In addition, chloride anions might also be incorporated in PPy, which contributes to the increase in electrical conductivity [40,41,42,43,44] (Figure 8).

In our study, a decrease in electrical resistance of the PPy-coated PA66 fabric due to exposure to HCl can be observed, as shown in Figure 9. For instance, a decrease in resistance between 73 and 79% can be observed within the first minute of exposure. The resistance decreases with increasing exposure time due to the absorption of a higher amount of HCl to the substrate. As a consequence, a higher level of protonation of the polymer is achieved, which then leads to an increase in conductivity (Figure 9a). The highest decrease in resistance is observed within the first minute, followed by the resistance levels. The adsorption of HCl reaches an equilibrium state, and the protonation of PPy stabilises. For higher HCl concentrations, the resistance levels after a shorter exposure time are compared to experiments with lower HCl concentrations in the atmosphere, which is due to faster adsorption during the initial phase [43,45].

With personal protective equipment, the sensor must react within a short period of time so that workers are not exposed to the hazardous environment for too long. This is why the results were extracted for the first three to five minutes. Three readings in the interval of 3–5 min were used to assess the response and concentration dependency of the sensor. Figure 9b shows the correlation between HCl concentration and resistance for the time interval of 3–5 min.

Furthermore, we studied the recovery properties of the sensor. For this, we removed the sensor from the HCl source and already observed an increase in resistance within the first minutes. The full recovery of a sensor that was exposed to the HCl solutions for 24 h was achieved after 100 h. A full recovery was defined as an increase in resistance to the initial level of the sensor. The full recovery of the sensor with time also confirms that possible corrosion of the Cu film did not disturb its functionality.

According to the literature back in the 1980s [43,45,46], the protonation of PPy by acidic treatment converts the non-conducting PPy to conducting PPy salt, and the reverse process is induced by alkali treatment, which leads to deprotonation of PPy (Figure 8). The results are in accordance with later studies of deprotonation/reprotonation of PPy [42,47]. However, it has also been reported that each cycle slightly deteriorates the conductivity of the PPy layer [42]. Such phenomenon has to be analysed in more detail in future studies for our samples in order to predict any loss in functionality as a function of usage time under relevant application conditions. 

#### 3.2.3. Polypyrrole Coated Polyamide 66 Fabric as Ammonia Sensor

The mechanism for reversible ammonia adsorption has been studied by several authors [11,48,49,50,51]. Gustafsson et al., for instance, proposed the attack of ammonia on acidic protons in polypyrrole. As a result, the positive charge density of the polymer and the conductance decreased [49]. Carquigny et al. proposed an electron loss of the electron pair of nitrogen (1), which is followed by an electron transfer between ammonia and the polymer (2) (Figure 10), which leads to a decrease in the positive charge density of the polymer [11]. After the adsorption of NH_3_, the conductance of the polymer decreased. In this publication, the authors proposed that ammonia is adsorbed onto the PPy surface, forming NH_3_^+◦^ radical groups. The proposed mechanism is supported by other research work investigating ammonia detection with the use of PPy-based gas sensors [48,52]. Scott et al. postulated a partial reaction mechanism for ammonia oxidation via an amine oxide on PPy in the presence of hydrogen peroxide as an oxidising agent [51]. 

As shown in Figure 11, the resistance of our sensors increases as a result of ammonia exposure (Figure 11a). The resistance increases within the first minute between 31 and 88%. With the increasing treatment time, the resistance increases due to higher adsorption of ammonia molecules and, thus, higher loss in the positive charge density. The highest increase in resistance can be observed within the first two minutes. Similar to the observations for HCl adsorption on PPy, the resistance levels at higher treatment durations. This might be again due to the saturation of the polymer with ammonia. Furthermore, a stronger increase in resistance can be observed for higher NH_3_ concentrations. Figure 11b shows the correlation between NH_3_ concentration and resistance for the time interval of 3–5 min. After 3 min exposure time (Figure 11b), the resistance shows an increase of 66, 83 and 100% for a NH_3_ concentration of 100, 200 and 400 ppm, respectively. Higher NH_3_ concentrations lead to higher adsorption of ammonia and, thus, a stronger loss in the positive charge density of PPy. 

Regarding the possible reuse of the sensor, a recovery could not be observed for the different NH_3_ concentrations after six hours of exposure time. This might be due to the irreversible effects of ammonia adsorption, such as the incorporation of ammonia into the PPy polymer [49]. 

It appears that either time or concentrations were too high to induce reversible adsorption of ammonia into PPy [42]. However, an exposure time of three to five minutes might lead to a reversible adsorption of ammonia and a reuse of the sensor surface. According to Prokes et al., the change in the conductivity of PPy by deprotonation/reprotonation is a reversible process. However, the slight decrease in conductivity after each cycle is likely due to the hydrogen bonds, which are created between the imine nitrogen of the deprotonated pyrrole ring and the NH group of the non-deprotonated pyrrole ring. Such interaction decreases the ability of imine nitrogen to be re-protonated [42]. The reversibility and, thus, reuse ability of PPy-coated textile for NH_3_ detection still needs to be investigated further in future work.

The functionality of the PPy sensor for gaseous HCl and NH_3_ was tested above aqueous solutions of appropriate concentration (Table 1 and Table 2). Remarkably, the presence of HCl led to a reduction in resistance, while an increase in resistance was observed in the presence of NH_3_. The flexible sensor thus also allows us to distinguish between these two hazardous gases. Possible interference of water vapour on the signal development of the sensor for different gas concentrations of HCl or NH_3_ can be neglected as this interference would either reduce or increase the resistance of the PPy sensor. As the signal of a PPy sensor might be influenced by larger differences in atmospheric moisture content, appropriate compensation should be considered to rule this effect out in future practical applications.

## 4. Conclusions

In this study, we demonstrated the potential of flexible PPy thin-film coated textiles as material for manufacturing sensors and suitable for integration into garments to produce functional e-textiles. We presented a simple synthesis that generates thin PPy layers on PA66 fabric. The high flexibility of the sensing element allows for their integration into garments. The sensor elements are multifunctional and can indicate temperature and be used as gas sensors for the detection of corrosive gases such as HCl or NH_3_. In the presence of 20–100 ppm gaseous HCL, a proportional reduction in resistance of the sensing element is observed, while 100–400 ppm NH_3_ causes a concentration-dependent increase in resistance. The direction of the resistance change in the sensor element allows for distinguishing between gaseous HCl and NH_3_. The change in resistance also allows a quantification of the respective gas concentration. When the PPy sensor element is covered by an impermeable thin layer, the sensor indicates temperature only. By combining HCl and NH_3_ gas-sensitive PPy sensor elements with a fully covered PPy sensor, the combination of devices allows for indicating both corrosive gases and temperature. The results of the temperature sensor can be used for temperature correction of the gas sensor signal. The sensors are of high interest to be integrated into personal protection cloth for workers in various industries to be alerted in case of danger. The simple and cheap manufacturing process of the sensor elements will allow for their design as disposable devices that can be replaced easily after an alert. Future work will address an improvement of the adhesion of the PPy layer, e.g., by activation of the PA66 fabric surface by complexation, in situ polymer grafting, and plasma treatment to enhance stability against wash ageing and mechanical abrasion [38,53,54]. The aspect of functional ageing of the coated fabrics [55] and the durable contacting of the sensor will also be addressed in future studies.

## Figures and Tables

**Figure 1 sensors-24-01387-f001:**
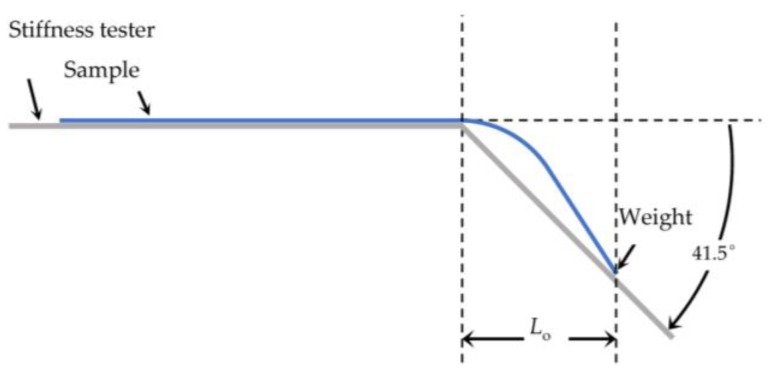
Setup for flexural rigidity measurement.

**Figure 2 sensors-24-01387-f002:**
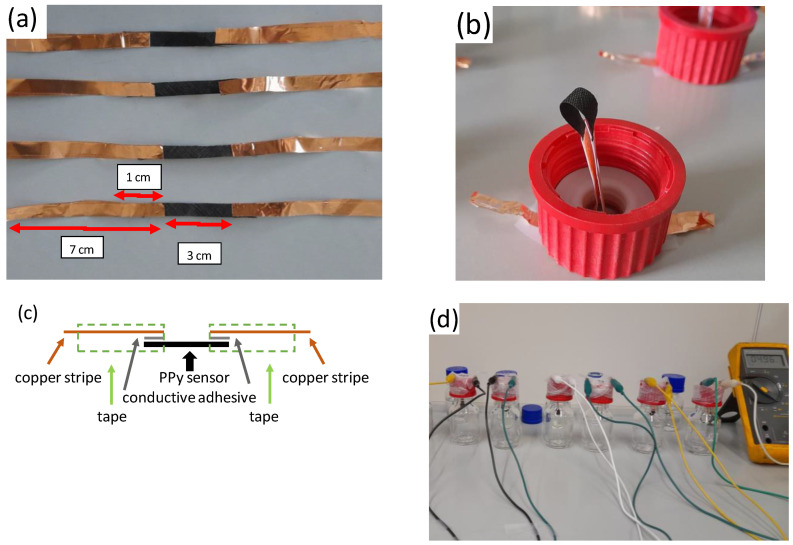
Experimental setup for the analysis of HCl and NH_3_ sensing properties of PPy-coated PA66 (explanations in the text). (**a**) sensor stripes contacted with copper foil, (**b**) installation of sensor element in closing cap, (**c**) schematic drawing of sensor element, (**d**) full set-up of gas sensors.

**Figure 3 sensors-24-01387-f003:**
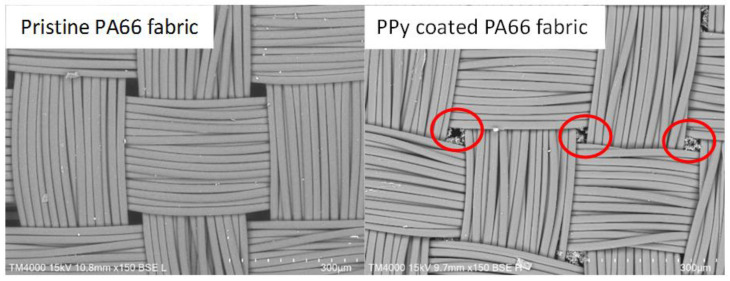
Top views of pristine PA66 and PPy-coated PA66 fabric (red circles indicate deposited PPy in free area between yarns).

**Figure 4 sensors-24-01387-f004:**
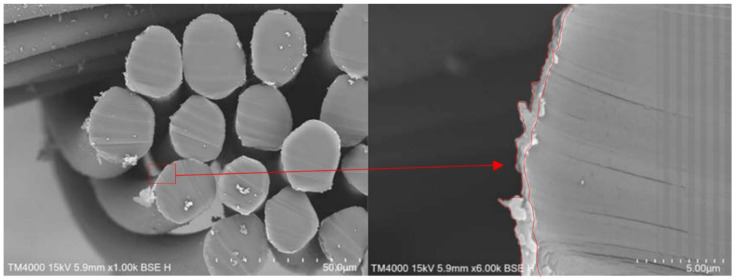
Cross-sections of PPy-coated PA66 fabrics (red arrow indicates magnified section).

**Figure 5 sensors-24-01387-f005:**
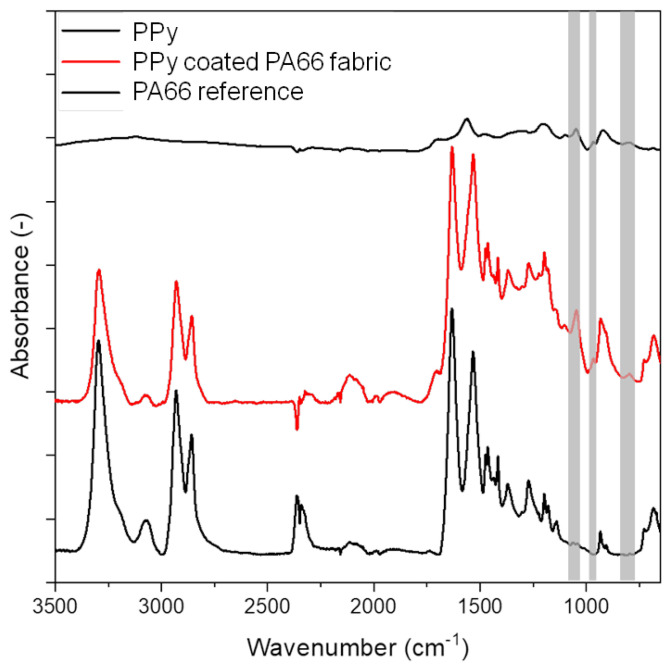
FTIR-ATR spectrum of PA66 and PPy coated PA66 fabric and PPy.

**Figure 6 sensors-24-01387-f006:**
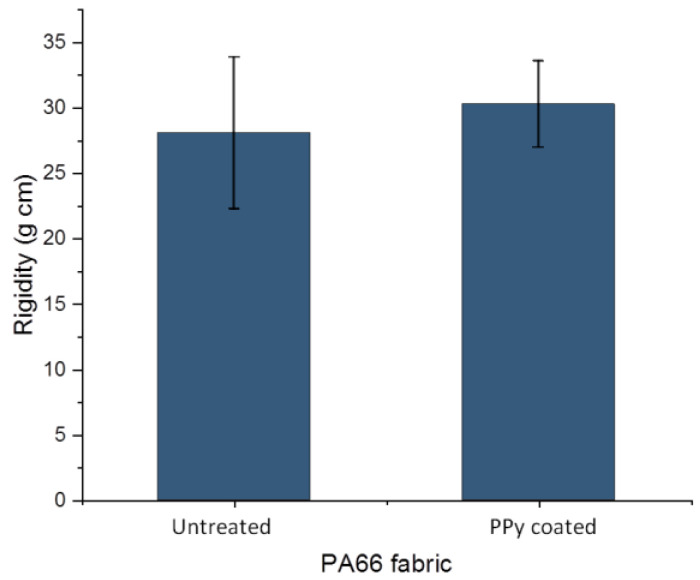
Flexural rigidity of PA66 and PPy-coated PA66 fabric.

**Figure 7 sensors-24-01387-f007:**
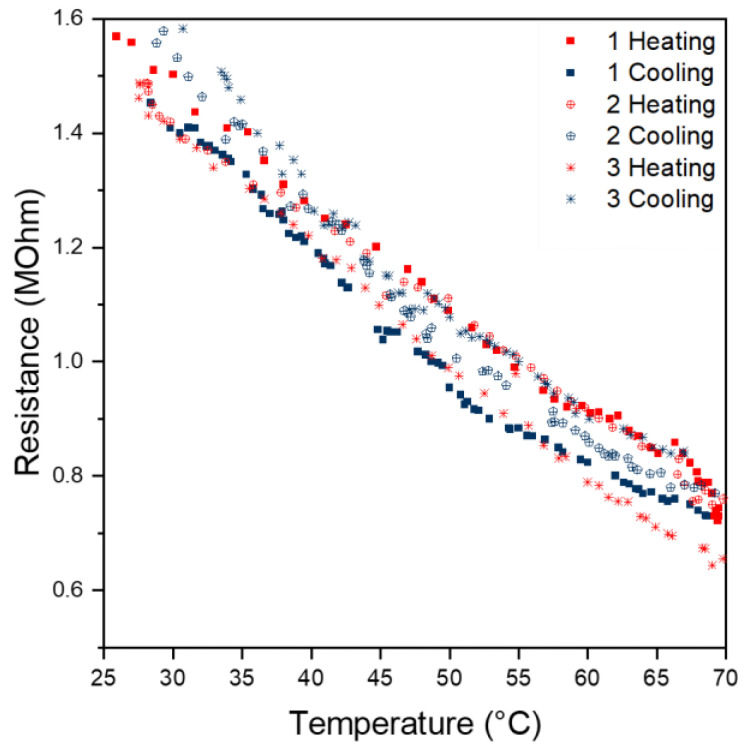
Correlation between time and resistance.

**Figure 8 sensors-24-01387-f008:**
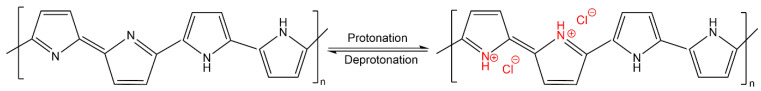
Mechanism for the adsorption of HCl on PPy according to [42].

**Figure 9 sensors-24-01387-f009:**
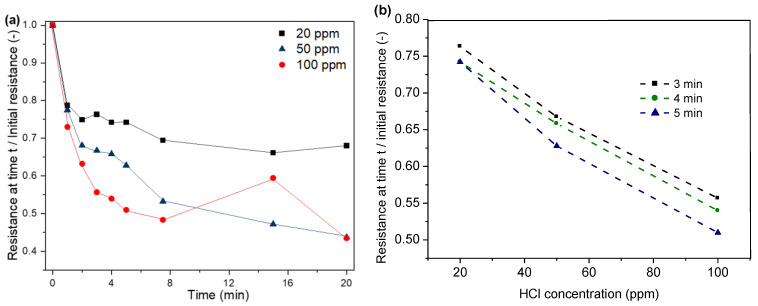
Correlation between time (**a**), HCl concentration (**b**) and normalised resistance.

**Figure 10 sensors-24-01387-f010:**
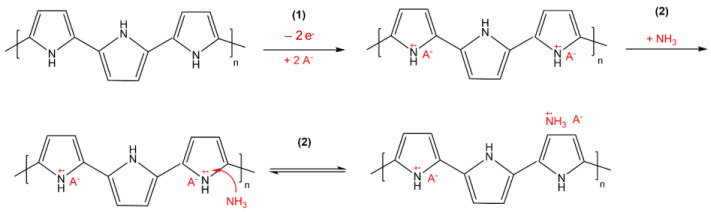
Mechanism for the adsorption of NH3 on PPy according to [11,20]; (1) electron loss of electron pair of nitrogen; (2) electron transfer between ammonia and polymer.

**Figure 11 sensors-24-01387-f011:**
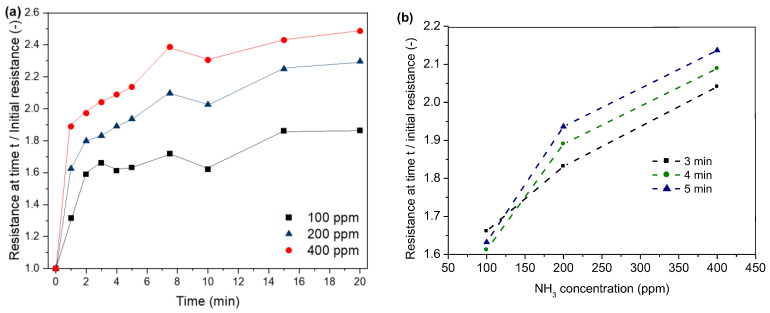
Correlation between time (**a**), NH_3_ concentration (**b**) and normalised resistance.

**Table 1 sensors-24-01387-t001:** Correlation between vapour pressure (20 °C) and concentration of HCl solution according to [29,31].

Concentration ppm	Vapour Pressure Pa	Concentration wt%
0	0	0
20	1.61	12.8
50	4.02	15.1
100	8.05	16.6

**Table 2 sensors-24-01387-t002:** Correlation between vapour pressure (20 °C) and concentration of NH3 solution according to [30,32].

Concentration ppm	Vapour Pressure Pa	Concentration wt%
0	0	0
100	17.4	0.40
200	34.8	0.42
400	69.8	0.46

## Data Availability

The data that support the findings of this study are available from the corresponding author upon reasonable request.

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
