# Peer review of "Multifunctional Polypyrrole-Based Textile Sensors for Integration into Personal Protection Equipment"

_sensors, 2024, doi:10.3390/s24051387_

Round 1

Reviewer 1 Report

Comments and Suggestions for Authors

The presented manuscript has some interest, as it describes the possible use of clothing fabrics as sensor elements. However, the manuscript has a number of disadvantages. The main disadvantage is that the presented study is incomplete. The authors write about this in the section "Conclusions".

To understand the processes taking place, it is also necessary to conduct experiments with water vapor. The electrical conductivity of the polypyroll can also change during the adsorption of water molecules. In addition, it is unclear how the contact of the sensory tissue with the copper tape was carried out. The authors don't write about it anywhere. And this is important during the course of adsorption processes. Copper can change its surface properties under the action of ammonia or HCl molecules. The response shown in Figures 9 and 11 may be related to a change in the contact resistance at the "fabric with pPy – copper" boundary.

There are also other comments on the manuscript.

L50-51. The phrase needs to be corrected. The surface area is usually defined per unit mass.

L51. What exchange of substances are we talking about here?

L56-58. This is a replay.

L101 and Fig.1. The Lb size is not shown in Figure 1.

L107, 125,127,129, 202,208. The numbering of the equations is indicated without the word "Equation".

L122-123. This is an inaccurate definition, since the molar masses of air, HCl and NH3 have different values. The presented estimate leads to an error in determining the concentration of 25%-42%.

Table 1 and 2. At what temperature are the saturated vapor pressure parameters specified? In addition, the dimension of the quantities must be specified in the international SI system of units. References to literature should also be indicated more modern. For example.......

L75 and 138. Different sizes of fabric samples are indicated in these places.

Fig. 2 . In the drawing, you need to specify where the sensor is located. Under the picture, it is necessary to explain what is shown in Fig. a), b) and c), or write the phrase "(explanations in the text)". When describing the drawing, it is necessary to indicate how the touch tissue was contacted with the copper tape. The authors don't write about it anywhere. This is important in the course of adsorption processes. It is desirable to show the sensor drawing in the section.

L186-187. It is better to write here that “flexibility" changes slightly.

L319-320, 322-330. The phrases are not the conclusion of this manuscript.

L436-437.  Misprint.

Reviewer 2 Report

Comments and Suggestions for Authors

The manuscript entitled "Multifunctional polypyrrole based textile sensors towards the integration in personal protection equipment” discusses on the facile preparation of PPy coated fabric for rapid response against temperature, acidic and ammonia exposure. The study sounds interesting, simple and viable, however, the following concerns needs to be addressed before considering the manuscript.

Abstrat

1.        The concluding part of abstract is not convincing. Authors must emphasize what has been established in the present work. Future recommendations can be included in the conclusion.

Introduction

2.        Page 1, Line 22-23: Similar sentence found in the abstract. Rephrase to make it different from the abstract, otherwise, it would be a repetition only.

3.        The polypyrrole usually abbreviated as PPy NOT pPY. Please revise throughout the manuscript.

4.        What’s the importance of selecting PA66 as the fabric substrate? Is it widely used by the workers (i.e. firefighters) who are in the need of these sensors? Justify.

5.        Why the ammonia was tested at an elevated concentration range? Is it a norm?

6.        Latest references on ammonia sensing must be included such as: https://doi.org/10.1016/j.ijbiomac.2023.124079

7.        The last paragraph of Introduction must emphasize on the significance of the present work.

Methods

8.        Page 2, Line 79: 0.04 m pyrrole? Do you mean 0.04M?

9.        Typically conducting polymers’ monomers such as aniline and pyrrole requires acidic environment (pH<2) to get initiated – pyrrole cation. However, in the present work – pyrrole was just dissolved in water? Could you provide a reference for your method? Authors also needs to be specific what type of water was used? i.e. distilled water vs deionized water.

Results & Discussion

10.   Since the fabric was immersed in the reaction media, why did PPy does not coat fabric surface? And it was assumed to be found on the free areas only?

11.   Elaborate the formation of new peaks? Which functional groups do they indicate?

12.   Equation (5) was not mentioned in the text.

13.   Usually the correlation of coefficients will be reported 0.99xx, do not convert it into %. Furthermore, authors’ R2 were calculated based on which component? – there are 3 heating stages and 3 cooling stages.

14.   Fig 9 a – why the resistance increased at 16th min? It doesn’t look a saturation point.

15.   Fig 9 b – why authors studied the calibration based 3 points only? Is it sufficient? Any replicates involved?

16.   Page 9, Line 261: very old references were reported. There are many ongoing PPy based ammonia sensors reported recently. For instance, check this paper: https://periodicos.ufms.br/index.php/orbital/article/view/17875, cite more recent literatures.

17.   Fig 11 b – why the response not calculated as normalized resistance? No linearity found for ammonia detection.

Comments on the Quality of English Language

Minor revision on language is required.

Reviewer 3 Report

Comments and Suggestions for Authors

The manuscript has proposed a novel approach to generate thin, homogeneous Polypyrrole layers on flexible textile fabrics which can act as sensors to detect temperature and gas. Through this manuscript, they have demonstrated the feasibility of preparing temperature and gas sensors by coating a polypyrrole thin layer on a flexible fabric. Therefore, I would recommend this work for publication, provided that the authors address the following comments sufficiently:

1.      In Figure2, the authors performed experiments at different concentrations of NH3 and HCl, used to evaluate the performance of the sensors. However, although the authors claim a relationship between concentration and vapor pressure, there is no validation of the standard measuring equipment. On the other hand, figure2(a) have formatting problems. Therefore, it is recommended that the authors use professional detection equipment in the experiments to monitor concentration changes while changing the errors in figure2(a).

2.      In Figure7, the author deduced the temperature and resistance relations and the curve according to a series of formulas. However, there are no relevant experiments in the paper to prove the resistance and temperature relations given by the authors. Therefore, it is recommended that the author can supplement the experiment with the resistance changing with the temperature in the paper.

3.      The authors mentioned the whole recovery of the sensors that were exposed to the HCl solutions for 24 h was achieved after 100 hours. Based on this the authors believed that the sensor could be reused, however, the authors did not verify it. Therefore, it is recommended that the authors can supplement the subsequent experiments and test the performance of the fully recovered sensor to compare the performance difference of the same sensor in the two experiments.

4.      The authors mentioned that polypyrrole sensors can be applied in the field of personal protective equipment, where sensors can detect temperature and gas. However, the authors did not explore how human temperature affects the performance of sensors to detect gas. Therefore, it is recommended that the author explain how human temperature affects sensor performance and supplement relevant experiments.

Comments on the Quality of English Language

The expression can be further improved.

Round 2

Reviewer 1 Report

Comments and Suggestions for Authors

The authors of the manuscript responded in detail to the comments and made the necessary adjustments. But the question arose, why is the revised manuscript not presented by template Sensors?

I think that after this correction, the manuscript can be accepted for publication.

Reviewer 2 Report

Comments and Suggestions for Authors

Authors have carried out necessary revision based on my comments, thus I recommend for the publication of the present work.

Reviewer 3 Report

Comments and Suggestions for Authors

Accept